# Pathophysiological Insight into Fatty Acid-Binding Protein-4: Multifaced Roles in Reproduction, Pregnancy, and Offspring Health

**DOI:** 10.3390/ijms241612655

**Published:** 2023-08-10

**Authors:** Yue Shi, Chi-Chiu Wang, Liqun Wu, Yunqing Zhang, Aimin Xu, Yao Wang

**Affiliations:** 1The Second Clinical Medical School, Beijing University of Chinese Medicine, Beijing 100078, China; shiyuesia@foxmail.com (Y.S.); yunqzhang@foxmail.com (Y.Z.); 2Department of Obstetrics and Gynaecology, Faculty of Medicine, The Chinese University of Hong Kong, Hong Kong; ccwang@cuhk.edu.hk; 3Li Ka Shing Institute of Health Sciences, School of Biomedical Sciences, Chinese University of Hong Kong-Sichuan University Joint Laboratory in Reproductive Medicine, The Chinese University of Hong Kong, Hong Kong; 4Department of Pediatrics, Dongfang Hospital, Beijing University of Chinese Medicine, Beijing 100078, China; wulq1211@163.com; 5State Key Laboratory of Pharmaceutical Biotechnology, The University of Hong Kong, Hong Kong; amxu@hku.hk; 6Department of Medicine, The University of Hong Kong, Hong Kong

**Keywords:** FABP4, metabolism, immune regulation, women reproduction, placenta, pregnancy complications, pregnancy outcomes, fetal–maternal health, offspring

## Abstract

Fatty acid-binding protein-4 (FABP4), commonly known as adipocyte-fatty acid-binding protein (A-FABP), is a pleiotropic adipokine that broadly affects immunity and metabolism. It has been increasingly recognized that FABP4 dysfunction is associated with various metabolic syndromes, including obesity, diabetes, cardiovascular diseases, and metabolic inflammation. However, its explicit roles within the context of women’s reproduction and pregnancy remain to be investigated. In this review, we collate recent studies probing the influence of FABP4 on female reproduction, pregnancy, and even fetal health. Elevated circulating FABP4 levels have been found to correlate with impaired reproductive function in women, such as polycystic ovary syndrome and endometriosis. Throughout pregnancy, FABP4 affects maternal–fetal interface homeostasis by affecting both glycolipid metabolism and immune tolerance, leading to adverse pregnancy outcomes, including miscarriage, gestational obesity, gestational diabetes, and preeclampsia. Moreover, maternal FABP4 levels exhibit a substantial linkage with the metabolic health of offspring. Herein, we discuss the emerging significance and potential application of FABP4 in reproduction and pregnancy health and delve into its underlying mechanism at molecular levels.

## 1. Introduction

Pregnancy constitutes a complex physiological process that engages robust immunological and metabolic adaptations [1]. These changes are precisely regulated by a myriad of hormones and cytokines, which are critical for ensuring a successful pregnancy [1]. The physiological dynamics include the establishment of the immune tolerance status for foreign alloantigen maintenance and endocrine alternation typified by insulin resistance [2]. Metabolic and immune maladaptation during pregnancy results in numerous complications, such as miscarriage, gestational diabetes mellitus (GDM), preeclampsia (PE), and preterm birth, which subsequently affect maternal long-term morbidity and mortality [3]. Moreover, these complications carry substantial intergenerational consequences, predisposing offspring to metabolic diseases later in life [4,5,6]. Consequently, a comprehensive understanding of the mechanisms and factors that orchestrate physiological adaptions during pregnancy is important for the prevention of reproductive and gestational complications and the improvement of maternal and child health.

Fatty acid-binding protein 4 (FABP4), a 14–15 kDa protein belonging to the lipocalin family, is also known as adipocyte fatty acid-binding protein (A-FABP) or adipocyte protein 2 (aP2) [7]. As an adipose-derived hormone, FABP4 is primarily expressed in adipocytes as a cellular marker of adipocyte maturation [8] and is mainly involved in immunometabolism implications and obesity-associated tumors [9]. Though FABP4 is predominantly enriched in adipose tissue [10], recent studies have revealed its expression in other cell types, including macrophages [11], dendritic cells [12], endothelial cells [13], and placental trophoblast cells [14]. FABP4 is a master regulator of lipid homeostasis [15]. It binds to and transports intracellular fatty acids (FA) and other lipophilic molecules to influence lipid metabolism [16]. FABP4 serves as an upstream activator that augments the activities of peroxisome proliferator-activated receptor gamma (PPARγ) [17], which regulates adipogenesis and adipocyte differentiation [18], as well as hormone-sensitive lipase (HSL) and adipose triglyceride lipase (ATGL), which are all key factors promoting lipolysis and FA oxidation [19]. In immune cells, FABP4 expression is induced by inflammatory stimuli, such as lipopolysaccharide (LPS) and tumor necrosis factor-alpha (TNF-α) [20]. The LPS-activated c-Jun N-terminal kinase (JNK) pathway increases the phosphorylation of c-Jun in macrophages [21], which in turn binds to the activator protein 1 (AP-1) cis-element within the FABP4 gene promoter to enhance its transcription [21]. Increased FABP4 further potentiates LPS-elicited downstream activation, thereby creating a positive feedback loop to promote inflammation [21]. Collectively, FABP4 is a pleiotropic protein that regulates lipid storage, lipogenesis, and lipolysis [22,23] and plays a crucial role in the pathogenesis of metabolic inflammation.

Growing studies demonstrate that elevated circulating FABP4 levels correlate with an increased risk for obesity [24], type 2 diabetes [25], various cardiovascular diseases [26], and cancers [27] in a wide variety of populations. In pregnancy, increased serum FABP4 concentrations are observed in pregnant women with maternal obesity [28,29], GDM [30,31,32,33,34,35,36,37,38,39], and PE [28,40,41,42]. However, its expression is detected in human placental trophoblast cells and is dispensable for maternal–fetal exchange and fetal growth [43]. Emerging evidence suggests that FABP4 performs a vital role at the fetoplacental interface, involving maternal–fetal immune tolerance and lipid transportation [13,14,44,45,46,47,48,49,50]. Thus, the specific function of FABP4 during pregnancy warrants further discussion.

In this review, we collate recent animal and clinical studies on the role of FABP4 in pregnancy and its implications for maternal and fetal health. We first summarize the current understanding of FABP4 in various reproductive disorders, pregnancy complications, and fetal health. We further introduce the molecular signals of FABP4 and the expression and regulation of FABP4 during pregnancy and review the potential mechanisms by which FABP4 may affect placental development and function, as well as fetal programming of metabolic health. Lastly, we explore the potential of FABP4 as a biomarker and therapeutic target for improving pregnancy outcomes and intergenerational health. This review delivers updated insights into the occurrence of reproduction dysfunction and gestational diseases via FABP4. 

## 2. The Role of FABP4 on Women’s Reproductive Health

### 2.1. Polycystic Ovary Syndrome (PCOS)

Polycystic ovary syndrome (PCOS), characterized by excess androgen and polycystic ovaries [51], is the most common endocrine dysfunction affecting 4% to 20% of women of reproductive age [52]. It is closely associated with the development of infertility and the risk of metabolic comorbidities, such as diabetes in women [51]. Wang et al. showed FABP4 gene polymorphism *rs3834363* was significantly associated with the development of PCOS [53], implying that FABP4 is a potential target gene for the etiology of PCOS. Compared to healthy subjects, FABP4 expression levels are increased in granulosa cells of patients with PCOS [54]. Numerous studies have reported that serum FABP4 levels were higher in PCOS patients than in controls and are associated with clinical phenotypes, including body mass index (BMI) and homeostasis model assessment-insulin resistance (HOMA-IR) [55,56], Interestingly, changes in circulating FABP4 levels were also related to the therapeutic outcomes of PCOS. Lolanda et al. showed that serum FAPB4 levels were lower in the metformin-treated group compared to untreated subjects, indicating that FAPB4 may implicate in the development of PCOS [57]. Therefore, serum FABP4 level may be a discriminative marker for predicting PCOS and subsequent metabolic syndrome.

### 2.2. Endometriosis

Endometriosis is featured by the deposition and proliferation of endometrial cells or tissues outside the uterine cavity, which poses another major risk factor for women’s reproduction [58]. Eyster et al. showed that FABP4 gene expression was significantly up-regulated in ectopic compared with the eutopic endometrium in women [59], which may be attributed to the infiltration and activation of M1-type macrophages in ectopic endometrial sites [60]. Conversely, the downregulation of FABP4 expression by miR-455 treatment significantly reduced oxidative stress and apoptosis in a human endometrial stromal cell line (HESCs) [61], suggesting the potential role of FABP4 in the development of endometriosis by inducing oxidative stress. In addition to the direct effects of FABP4 on endometriosis by targeting endometrial cells, a clinical study found that adipose tissues adjacent to the endometriotic lesions also exhibited higher FABP4 and vascular endothelial growth factor (VEGF) expression levels [60], suggesting FABP4 may promote fat fibrosis to induce endometriosis. These findings indicate that FABP4 may contribute to endometriosis by affecting inflammation and metabolic disruption both systemically and locally. However, no clinical study has yet determined the circulating FABP4 levels in patients with endometriosis. Further investigation is needed to evaluate whether FABP4 serves as a potential biomarker for this disease. 

## 3. The Role of FABP4 on Pregnancy Health

### 3.1. The Molecular Mechanism of FABP4 Regulating Placental Function

The placenta is an integral organ in the maintenance and development of pregnancy, whereas placental dysfunction results in serious complications for both mother and fetus [62]. Within the placenta, FABP4 expression was first identified in the placental labyrinth, especially in endothelial cells [13], and has also been detected in human epithelial cells of the uterine endometrium [63]. It profoundly modulates immune hemostasis, lipid metabolism, and fetal development by targeting the placenta throughout gestation. 

The establishment and maintenance of pregnancy depend on the interaction between the embryo and the maternal uterine endometrium [64]. In the uterine endometrium, the expression of FABP4 is site-specific and time-dependent during gestation. Previous animal studies found that FABP4 is primarily expressed in the labyrinthine layer, and its expression reaches the maximum level at 14.5–16.5 gestational days [13]. In humans, FABP4 presents in the trophoblast layer and villous endothelial cells of the placenta [14]. Further study also showed FABP4 is notably expressed in endometrial epithelial cells during proliferative and secretory phases and in stromal cells in the secretory phase, where it regulates the proliferation, migration, and invasion of epithelial cells in the endometrium [63]. 

Of note, FABP4 expression has also been detected in various immune cells resident in the placenta, including macrophages [65], dendritic cells [12,66], and natural killer (NK) cells [12], which are instrumental in shaping the immune environment at the maternal–fetal interface. In macrophages, increased FABP4 has been shown to modulate macrophage polarization by promoting the M1 pro-inflammatory phenotype, and decreased FABP4 levels favors the M2 anti-inflammatory phenotype [11,46]. It triggers macrophage inflammation through activation of the Janus kinase 2/signal transducer and activator of the transcription 2 (JAK2/STAT2) pathway [47] and may promote macrophage inflammation via a reduced reactive oxygen species (ROS)-dependent mechanism [48]. In contrast, the absence of FABP4 in macrophages enhances the expression of sirtuin 3 (SIRT3) [67] and ROS production [68] for anti-inflammatory effects. Moreover, FABP4 influences NK cell activity and cytotoxicity by mediating lipid metabolism [69]. NK cells collected from an obese condition with a lipotoxic environment exhibited increased FABP4 expression levels aligned with lipid accumulation [69]. Further study showed that upregulation of FABP4 expression impaired NK cell function in the release of interferon-γ (IFN-γ) [70], which is a vital cytokine for pregnancy success [49]. Overall, given the importance of placenta–resident immune cells in pregnancy, these regulatory effects of FABP4 on immune cells have significant implications for immune surveillance and tolerance at the maternal–fetal interface, thereby potentially affecting placental functionality and development.

Furthermore, FABPs are essential for active FA transport, metabolism, and gene expression in the placenta for substantial intergenerational consequences [44]. A human study found FABP4 has a synergistic effect on placental lipid transport and accumulation at the maternal–placenta interface and endothelial layer to maintain high FA uptake rates into the placenta [71]. An in vitro study using human trophoblasts demonstrated FABP4 is expressed in both the trophoblast layer and fetal capillaries of placental villi [14], which is in charge of transplacental transport of lipids from the maternal blood into the fetal circulation through the syncytiotrophoblast and capillary endothelial cells [45]. Primary human trophoblasts (PHT) exposed to selective FABP4 inhibitor BMS309403 displayed decreased FA-induced FABP4 protein expression, reduced accumulation of lipid droplets, and abolished FA-induced triglycerides in trophoblasts [14]. Similarly, FABP4 knockdown by siRNA also attenuated lipid droplet buildup in PHT cells [14]. These results indicate that FABP4 plays a pivotal role in the uptake and accumulation of lipids in human trophoblasts. Moreover, the overexpression of FABP4 in pregnant rats and trophoblast cells increased transplacental transport of docosahexaenoic acid (22:6 n-3, DHA) without significant placental DHA accumulation [72], indicating FABP4 may play a more crucial role in FA transport than accumulation in placental lipid metabolism. However, current evidence proposes FABP4 might be dispensable for fetoplacental lipid transport. Minimal changes in weight and morphology were observed between FABP4^−/−^ fetuses and wild type littermates, and similarities were noted in both maternal serum and fetal hepatic total cholesterol (TC) levels among the different genotypes [13], indicating that FABP4 deficiency did not impact the supply of TC to the developing fetus and predisposing offspring to metabolic diseases later in life. Further studies are still warranted to determine the impact of FABP4 in fetal lipid supply.

Taken together, FABP4 plays a crucial role in pregnancy establishment and maintenance, acting in different aspects, such as lipid metabolism, immune modulation, and fetal development (Figure 1). However, despite our understanding of its involvement, further research is necessary to elucidate the detailed mechanisms of FABP4 in maternal–fetal interface regulation and its potential as a therapeutic target for reproductive and pregnancy-related disorders as shown below. 

### 3.2. Implantation Failure and Pregnancy Loss

Maternal inflammation constitutes a significant risk factor for pregnancy loss [43]. During embryo implantation, immune cells, such as macrophages and NK cells, aggregate and activate at the maternal–fetal interface to initiate immune responses [50,75]. Intriguingly, FABP4 has been identified in multiple studies as a crucial regulator of immune cells. It is highly expressed in activated NK cells to support mitochondrial metabolism [70]. In macrophages, increased FABP4 expression leads to the macrophage polarization from an anti-inflammatory status (M2) to a pro-inflammatory phenotype (M1), which consequently disrupts the immune homeostasis at the maternal–fetal interface and leads to failures in embryo implantation and decidualization [76]. Consistently, a recent study by Yang et al. determined serum FAPB4 levels was significantly higher in patients experiencing miscarriage at 8–12 weeks of gestation compared to age- and BMI-matched healthy pregnant women (Table 1), positing that serum FABP4 levels may be a marker in predicting miscarriage [77]. 

FABP4 plays a vital role in embryo implantation, which has been confirmed by cell and mouse experiments. FABP4 silencing in the endometrial cell line significantly reduced the number of trophoblast spheroids adhered onto endometrial cells compared with scramble [43]. Moreover, FABP4 inhibitor treatment significantly impaired tube formation mediated by VEGF, DHA, and leptin, indicating the differential role of FABP4 in FA and angiogenic growth factors-mediated tube formation in the first-trimester trophoblast cells [78]. Mice administrated with FABP4 siRNA on the first day of pregnancy showed a decrease in the number of implanted embryos [43]. All these findings underscore that FABP4 is essential in the establishment and maintenance of pregnancy, and subsequently may be involved in the pathogenesis of implantation failure.

FABP4 may also be important for successful gestation during the post-implantation period, which provides essential nutrients to facilitate proper placental development [79]. Inhibiting nutrient sensors, including mechanistic targets of mechanistic target of rapamycin (mTOR) and PPARs, cause embryo loss, growth delay, and placental growth impairments in pregnant mice models [80]. An in vivo study demonstrated that inhibition of mTOR signaling results in embryo resorption and diminished FAPB4 expression in the decidual [80]. Likewise, blocking PPARγ or PPARδ pathway by chemical inhibitors also triggers fetoplacental growth dysfunction aligned with FABP4 downregulation [80], suggesting that FABP4 may play an important role in sustaining post-implantation growth.

Therefore, FABP4 may contribute to early pregnancy loss by affecting immunological and metabolic adaption. However, the question remains whether FABP4 represents a detrimental factor or a protective mechanism for miscarriage since the results of in vivo and in vitro studies were not consistent with the clinical finding. Additional investigations with FABP4 deficient mice model and clinical observations are necessary to definitively establish its specific roles in pregnancy maintenance.

### 3.3. Maternal Obesity

Obesity, a well-known risk factor for poor metabolic health, exhibits particularly detrimental effects during pregnancy. In pregnancy, maternal obesity is defined as being overweight (25.0–29.9 kg/m^2^) and obese (≥30 kg/m^2^) [81]. It is closely associated with the occurrence of gestational disorders, including PE and GDM [82]. Importantly, maternal obesity poses deleterious impacts on the offspring’s metabolic health later in life [83]. Extensive studies have proven that fetuses exposed to maternal obesity have a significantly elevated risk to develop obesity, hypertension, hyperglycemia, insulin resistance, hyperlipidemia, and non-alcoholic fatty-liver disease in their progeny including [84,85].

Circulating FABP4 levels are widely associated with obesity risk factors such as triglyceride, cholesterol, and leptin levels in humans and rodents [86]. Consistently, lines of evidence also revealed that serum FABP4 levels are positively associated with maternal BMI [28], especially in the subgroup with BMI surpassing 25 [29]. Moreover, maternal circulating FABP4 levels are also negatively associated with serum adiponectin levels, which is a metabolically beneficial hormone [29]. These findings underscore that elevated FABP4 levels are intricately linked to adiposity gain during pregnancy, thus suggesting that FABP4 levels could serve as a predictive indicator of the degree of maternal obesity.

### 3.4. Gestational Diabetes Mellitus (GDM)

GDM, characterized by the onset or first-recognized glucose intolerance in pregnancy, is diagnosed based on International Association of Diabetes and Pregnancy Study Group (IADPSG) criteria: fasting glucose level of ≥5.1, and/or 2-h ≥ 8.5 mmol/L after a standardized oral glucose tolerance test (OGTT) [87]. It is the most common pregnancy complication affecting 14.0% of pregnant women worldwide [87], which increases the risk of multiple adverse outcomes in mothers and their offspring [88]. The positive association between elevated serum FABP4 level and glycemic dysfunction has been well recorded in a general population [89,90]. Similarly, the elevation in serum FABP4 level may indicate an increased risk in GDM. Accumulating clinical evidence also identified that FABP4 levels are positively associated with tumor necrosis factor (TNF)-α and interleukin (IL)-6 levels in the serum of GDM patients [91,92]. In line with this clinical observation, suppression of FABP4 using a chemical inhibitor (BMS309403) led to decreased TNF-α and IL-6 levels, ameliorating glucose metabolism, and insulin tolerance in the GDM mouse model [36]. Glucose stimulates FABP4 expression in trophoblast cells [93], which enhances lipolysis and exacerbates pregnant insulin resistance (IR) [31,35], especially in the first and second trimesters [39]. These findings indicate FABP4 may aggravate the symptoms of GDM and the activation of inflammatory pathways [29].

Interestingly, serum FABP4 levels are strongly associated with the severity of GDM and postpartum consequences (Table 1). A majority of clinical studies have reported higher serum FABP4 levels in women with GDM than in euglycemia pregnancies and positive correlations with biochemical parameters abnormality [31,33,35,36,37,38,39,94]. Only three studies reported no significant difference in serum FABP4 levels between groups with and without GDM [29,30,34]. A nested case-control study revealed that women in the upper tertial of FABP4 levels in the first and second trimesters had 5.3% and 44.7% higher risk of developing GDM respectively, compared to those in the lowest tertial [39]. A prospective cohort study found FABP4 levels increased remarkably in GDM groups from the second to the third trimester and were associated with a higher risk of GDM onset compared to other adipokines, such as leptin and retinol-binding protein 4 (RBP4) [32]. However, one study reported that the differences in maternal serum between GDM and euglycemic pregnancies appear to recede one week prior to delivery [30], which may be attributed to the increased accumulation of maternal adipose depot and natural progression in increased insulin resistance status during late pregnancy [82,95]. Therefore, FABP4 may serve as a valuable biomarker in the discrimination of at-risk GDM subjects, especially in the in mid-late pregnancy.

### 3.5. Preeclampsia (PE)

PE is a severe pregnancy-induced hypertensive condition diagnosed as systolic blood pressure ≥ 140 mmHg and diastolic blood pressure ≥ 90 mmHg after 20 weeks of gestation concomitant with proteinuria and edema [96]. It is associated with multisystem disorders in the later stage of gestation, thereby leading to maternal and neonatal morbidity and mortality [97]. A nested case-control study detected markedly increased maternal serum FABP4 levels at 8 to 13 weeks of gestation in subjects who were eventually diagnosed with preeclampsia, suggesting that FAPB4 is a potential predictive biomarker for PE [40]. In accordance, among pregnant women with type 1 diabetes, those who developed PE exhibited increased FAPB4 levels during the first and second trimesters [42]. Another prospective longitudinal cohort in women with type 1 diabetes also demonstrated a significantly higher serum FABP4 level throughout the gestational period in those who later developed PE [41] (Table 1). Animal study further corroborated this by showing that placental FAPB4 expression level was augmented in a preeclamptic rat model [98]. Importantly, overexpression of FAPB4 in trophoblast cells induced pro-inflammatory cytokines (e.g., IL-6 and TNF-α) secretion and triggered intracellular lipid accumulation [98], which are considered putative pathogeneses of preeclampsia [99,100]. Overall, these findings indicate that FAPB4 is a potential biomarker for the early detection of PE, and it may implicate the onset of PE by inducing placental inflammation. However, since PE is aligned with GDM and maternal obesity as aforementioned, which also contribute to the elevated serum FABP4 level. Therefore, whether circulating FABP4 level is an independent risk factor for PE should be further investigated by adjusting cofounding factors, e.g., BMI.

### 3.6. Preterm Birth

Premature birth occurs before the 37th week of pregnancy [101], accounting for 11% of all perinatal mortality and morbidity births worldwide [102]. Notably, preterm birth ranks as the leading cause of mortality among children under five years old [103] and predispose newborns to numerous early-age diseases as well as sensory impairment, learning disabilities, and respiratory illness in their later life [104].

FABP4 may be involved in premature delivery via immunologically generated processes in pregnant tissues, which mirrors the mechanism of pregnancy loss [12]. Higher serum FABP4 levels were measured in premature delivery in a clinical study [105], especially in very and extremely preterm infants (Table 1). Lines of evidence have demonstrated that FABP4 in cord blood [105] and serum of one-month infants [106] tends to be higher in preterm neonates than in neonates delivered at term, and this difference increases with a lower gestational age [105]. A cross-sectional study detected higher FABP4 levels in infants with gestational age less than 30 weeks compared to full-term infants [105]. Likewise, A prospective cohort study observed that preterm neonates with a mean gestational age of 32.8 weeks had slightly higher FABP4 levels compared with the full-term infants [106]. Therefore, there could be a positive correlation between serum FABP4 levels and the onset of preterm labor.

## 4. The Impact of FABP4 on Offspring’s Health

### 4.1. Neonatal Glucose Metabolism

Fetal glucose production is suppressed during pregnancy [107] and rapidly activated after delivery [108]. In neonates, FABP4 affects FA metabolism and plays a key role in neonatal glucose regulation [109].

As a glycemic regulating hormone, FABP4 intensifies the biological activity of glucagon and delivers free FA as an energy source to induce neonatal hepatic glucose production [108,109,110]. FABP4 levels in fetal circulation were found to surpass those in the maternal circulation, which further increased after birth [109]. Of note, FABP4 levels were mostly elevated in hypoglycemic neonates compared to healthy subjects [109]. Serum FABP4 levels in mouse neonates within 12 h from delivery inversely correlated with blood glucose, and the injection of recombinant FABP4 protein resulted in a rapid and significant increase in blood glucose levels and restored normoglycemia [109]. Animal studies using FABP4 knockout mice demonstrated that FABP4 deficiency disrupted the expression of multiple glucagon-regulated pathways, leading to impaired glycogen degradation in hypoglycemia [109]. Moreover, in the liver of 48-h-old FABP4 knockout mice, hepatic *G6pc* mRNA gene expression levels were remarkably increased, which is a critical gene regulating gluconeogenesis [109]. This indicates a potential role of FABP4 in the regulation of postnatal glucose metabolism in neonates.

### 4.2. Macrosomia and Low Birth Weight

Abnormal birth weight is a major risk factor for neonatal death and later-onset metabolic comorbidities [111]. Fetal growth restriction (FGR) is currently diagnosed as less than the 10th percentile of ultrasonography-estimated fetal weight (EFW) for the specific gestational age (GA) according to the latest standard [112], whereas macrosomia is characterized by birth weight exceeding 4000 g [113]. Since FABP4 mediates the transport of FA across the placenta into fetal circulation [14], it may influence fetal growth and neonatal size at birth by regulating placental function and fetal nutrient provision [114,115,116]. In low-birth-weight infants, FABP4 expression level in the adipose tissue was compensatory increased [117], especially in visceral fat [118,119]. In agreement with these results, a prospective cohort study confirmed a U-shaped relationship between serum FABP4 levels and birth weight in a healthy neonatal population [117]. However, in macrosomia cases, most of the studies reported increased FABP4 levels in placentas [120,121], except for a study that found no association between maternal serum FABP4 concentration and neonatal anthropometry [122]. Intriguingly, a cohort study found that FABP4 was significantly higher in the smaller twins compared with their normal-sized co-twins (Table 1) [123]. Collectively, current evidence supports the U-shaped relationship between FABP4 expression and fetal weight, underlining the importance of FABP4 in neonatal body weight and size.

**Table 1 ijms-24-12655-t001:** Summative table of clinical studies of FABP4 in pregnancy complications.

Year	Disease/ Condition	Sample Size (Patients vs. Control)	Region	Time of Sample Collection	FABP4 Levels (Patients vs. Control, ng/mL)	Measurement	Main Findings	Reference
2021	Preg-nancy loss	78/79	Xianyang, China	Serum; fasting, 8–12 wk of gestation	28.96 ± 19.14/ 19.64 ± 10.59 *p* < 0.001	ELISA	• Significantly elevated FABP4 levels in missed abortion than in control group independent of maternal age, BMI, or gestational age. • A discrimination ability of FABP4 levels with an AUROC of 0.70. • An improved distinguishing capacity (OR with 95% confidence intervals) of the FABP4 levels to 0.74 by cutoff value.	Yongkang Yang, et al. [77]
2011	GDM	98/86	Madrid, Spain	Serum; undescribed, <1 wk before delivery	17.7 ± 0.8/19.9 ± 1.0 *p* < 0.001	ELISA	• Significantly elevated FABP4 levels in GDM than in control group. • Positive correlations between FABP4 levels and neonatal fat mass, glycerol, and leptin in cord blood in GDM group (*p* < 0.05).	Henar Ortega- Senovilla, et al. [30]
2015	GDM	30/30	Hohhot, China	Plasma; fasting, 24–28 wk of gestation	14.7 ± 2.5/12.0 ± 0.7 *p* < 0.01	ELISA	• Elevated FABP4 levels in GDM than in control group.	Yuanyuan Li, et al. [31]
2016	GDM	40/240	Guangzhou, China	Serum;		ELISA	• Elevated FABP4 levels in GDM than in control group. • An increase of FABP4 levels from the second to third trimester in GDM group. • GDM was an independent factor of FABP4 levels (*p* < 0.05).	Ying Zhang, et al. [32]
				undescribed; 24–28 wk of gestation	32.35 ± 3.06/ 22.01 ± 2.00 *p* < 0.01			
				undescribed, ≥37 wk of gestation	36.47 ± 4.00/21.79 ± 1.32 *p* < 0.01			
2017	GDM	52/71	multiple center, Australia	Blood;		multiple technology	• A positive correlation between FABP4 levels and IR in early pregnancy, but no longer significant at 28 wk of gestation.	Kym J Guelfi, et al. [34]
				undescribed, 12–14 wk of gestation	19.4 (10.2–35.1) /20.3 (11.1–29.0) *p* < 0.05			
				undescribed, 28 wk of gestation	24.6 (16.4–41.5) /21.8 (15.5–28.8) n.s.			
2017	GDM	135/1015	Haerbin and Beijing, China	Plasma; fasting, 24–28 wk of gestation,	23.9 (17.6–32.2)/16.8 (12.3–23.1) n.s.	ELISA	• An optimal cutoff value of FABP4 level for the subsequent development of GDM at 18.5 ng/mL, with a sensitivity of 81.8% and a specificity of 71.2%. • An increased risk of GDM associated with FABP4 levels ≥ 18.6 ng/mL.	Wenjun Tu, et al. [35]
2017	GDM	50/50	Guangzhou, China	Serum; undescribed, ≥37 wk of gestation	20 ± 10.38/10.50 ± 5.69 *p* < 0.05	ELISA	• Elevated FABP4 levels in GDM than in control group	Ying Zhang, et al. [33]
2020	GDM	54/55	Zibo, China	Serum; undescribed, 24–30 wk of gestation	5.46 ± 1.36/ 1.34 ± 0.44 *p* < 0.001	ELISA	• Significantly elevated FABP4 levels in GDM than in control group. • Significantly positive correlations between TNF-α, IL-6 and FABP4 in GDM group (*p* < 0.001).	Bide Duan, et al. [36]
2020	GDM	107/214	multiple center, USA	Plasma;	(processed data)	ELISA	• A significantly positive correlation between GDM risk and FABP4 levels at 10–14 wk of gestation after adjusting (*p* < 0.0001).	Ellen C Francis, et al. [37]
				undescribed, 10–14 wk of gestation				
				fasting, 15–26 wk, 23–31 wk, 33–39 wk of gestation				
2020	GDM	60/50	Jinan, China	Serum; fasting, 24–28 wk of gestation	35.14 ± 11.39/21.53 ± 8.89 *p* < 0.001	ELISA	• Significantly elevated FABP4 levels in GDM than in control group. • Significantly positive correlations between TNF-α, IL-6 and FABP4 levels in GDM group (*p* < 0.001).	Xueling Wang, et al. [38]
2021	GDM	135/135	Beijing, China	Plasma;		ELISA	• Similar FABP4 levels in GDM and control groups. • A significantly positive correlation between FABP4 levels and IR (*p* < 0.001).	Chuyao Jin, et al. [39]
				undescribed, <14 wk of gestation	53.3 (33.1–93.2)/42.4 (32.6–63.8) n.s.			
				undescribed, 25–28 wk of gestation	53.8 (36.8–94.1)/41.6 (33.4–64.1) n.s.			
2021	GDM	22/18	Israel	Serum of cord blood; at birth	23.8 ± 16.9, umbilical artery; 22 ± 13.9, umbilical vein/15.2 ± 7.6 *p* < 0.05	ELISA	• Elevated FABP4 levels in infants born from GDM mother than in control group. • A further rise of neonatal FABP4 levels within the first 6 h of birth in all infants. • No correlation between FABP4 levels and birth weight in all infants. • An inverse correlation between FABP4 levels and blood glucose in all infants (*p* = 0.022).	Idit Ron, et al. [108]
2022	GDM	40/40	Tartu, Estonia	Serum; Fasting; 25–28 wk of gestation	80.30/94.03 n.s.	ELISA	• Similar FABP4 levels in GDM and control groups. • Significantly elevated FABP4 levels in women with BMI > 25 in GDM and control groups. (*p* < 0.001). • A significantly positive correlation between FABP4 levels and BMI in GDM and control groups (*p* < 0.001).	Tamara Vorobjova, et al. [29]
2008	PE	16/20	Leipzig, Germany	Serum; Fasting; <20 wk of gestation	24.5 ± 9.7/ 14.8 ± 7.1 *p* < 0.05	ELISA	• BMI and creatinine together explained 58% of the variation in FABP4 levels. • Elevated FABP4 levels in PE than in control group. • Elevated FABP4 level in patients with BMI > 25 than BMI < 25 (*p* < 0.05) in PE group.	Mathias Fasshaue, et al. [28]
2012	PE	22/72	Pittsburgh, PA, USA	Serum;		ELISA	• Elevated FABP4 levels in PE than in control group. • Similar magnitude of change of FABP4 levels in PE and control groups.	Christina M Scifres, et al. [40]
				Fasting; <13 wk of gestation	20.4 ± 12.3/ 10.1 ± 4.7 *p* < 0.05			
				Fasting; 24–28 wk of gestation	20.7 ± 11.7/ 9.9 ± 4.5 *p* < 0.05			
2014	PE	26/46	Helsinki, Finland	Serum;		ELISA	• Similar FABP4 levels in PE and control groups. • An inverse correlation between FABP4 levels and weight gain during pregnancy in women with pregnancy-induced hypertension (*p* = 0.02).	A L Tuuri, et al. [42]
				Fasting; 24 wk of gestation	20.1 (13.1–72.9)/ 19.2 (10.5–49.7) n.s.			
				Fasting; 32 wk of gestation	25.9 (12.4–213.7)/ 23.0 (13.2–59.5) n.s.			
2017	PE	23/19	Charleston, SC, USA	Plasma;		ELISA	• Elevated FABP4 levels in PE than in control group in 21.7 ± 1.4 wk and 31.4 ± 1.5 wk of gestation, both groups with GDM. • Progressive increase of FABP4 levels throughout pregnancy in all participates.	Clare B Kelly, et al. [41]
				Undescribed; 12.4 ± 1.8 wk of gestation	9.4 (7.9, 11.1)/ 7.6 (6.5, 9.0) n.s.			
				Undescribed; 21.7 ± 1.4 wk of gestation	11.1 (9.4, 13.1)/ 8.0 (6.8, 9.3) *p* < 0.01			
				Undescribed; 31.4 ± 1.5 wk of gestation	15.6 (13.4, 18.1)/ 10.2 (8.8, 11.7) *p* < 0.001			
2010	Pre-mature delivery	55/23	Athens, Greece	Serum; Undescribed; 31.9 ± 10.4 days of life	29.2 (14.5)/ 26.2 (8.9) n.s.	ELISA	• Similar FABP4 levels in preterm and full-term infants. • A positive correlation between FABP4 levels and TC in preterm and full-term infants (*p* < 0.05) • An independently and positive correlation between FABP4 levels and weight gain in preterm infants (*p* = 0.01).	Tania Siahanidou, et al. [106]
2017	Pre-mature delivery	144/217	Boston, MA, USA	Plasma of cord blood; at birth	48.2 (31.2~73.3)/ 35.8 (25.1~51.5) *p* < 0.01	ELISA	• Elevated FABP4 levels in preterm than full-term infants. • Elevated FABP4 levels in all infants than adults (*p* < 0.01). • Lower FABP4 levels in SGA than AGA group in full-term neonates (*p* = 0.05). • Elevated FABP4 levels in infants born from mothers with gestational hypertension or PE than from healthy mothers after controlling (*p* = 0.003).	Kyoung Eun Joung, et al. [105]
2021	Neonatal size	321	multiple center, USA	Plasma;	processed data	ELISA	• No correlation between FABP4 levels and neonatal anthropometry.	Ellen C Francis, et al. [122]
				Fasting; 15–26 wk of gestation				
				Random; 10–14 wk, 23–31 wk, 33–39 wk of gestation				
2019	IUGR	28/13	Leipzig, Germany	Plasma of cord blood; at birth	103.08 ± 80.61/ 49.63 ± 23.75 n.s.	ELISA	• Slightly elevated FABP4 levels in IUGR than in control group. • Elevated FABP4 levels in the smaller twins than in their larger co-twins (*p* = 0.028). • A significantly inverse correlation between FABP4 levels and birth weight and gestational age in all infants (*p* < 0.001).	Susanne Schrey-Petersen, et al. [123]
2022	Macro-somia	38/39	Wenzhou, China	Placental tissues; at birth	FABP4 mRNA relative expression	real-time PCR	• An increased mRNA expression of placental FABP4 in macrosomia than in control group (*p* < 0.05).	Li-Fang Ni, et al. [121]

All the studies are listed by the order of the year of publication and diseases. The studies published in the same year are listed by the order of initials of the first author’s name A-Z. ELISA, enzyme linked immunosorbent assay; FABP4, adipocyte fatty acid-binding protein; wk, week; BMI, body mass index; AUROC, area under the roc curve; OR, odds ratios; GDM, gestational diabetes mellitus; PE, preeclampsia; TC, total cholesterol; SGA, small-for-gestational age; AGA, appropriate for gestational age; IR, insulin resistance; IUGR, intra-uterine growth restriction; PCR, polymerase chain reaction. TNF, tumor necrosis factor; IL, interleukin. “Fasting” indicated the blood samples were collected in fasting status. “Undescribed” indicated the authors did not specify the time of blood sample collection. FABP4 levels were presented as mean ± S.D. or median with (interquartile range). n.s. indicated non-statistical significance. *p*-value was stated according to their own statistical method.

## 5. Influence of Environmental Factors on FABP4 during Pregnancy

Nutrition imbalance during pregnancy profoundly affects both maternal and fetal health. For instance, iron deficiency is implicated with anemic-associated maternal deaths [124] and spontaneous abortion [125]. Vitamin D (VD) deficiency is associated with GDM, miscarriage, and stillbirth [126]. Calcium deficiency is associated with PE [127] and pregnancy lactation-associated osteoporosis [128]. Moreover, dietary lipid intakes, such as n-3 polyunsaturated fatty acids (PUFAs) and long-chain polyunsaturated fatty acids (LCPUFAs), are indispensable in utero for the growing fetal brain and neurodevelopment [129,130].

Interestingly, growing evidence suggests the effects of inadequate micronutrients potentially through alterations in FABP4 expression. Maternal iron deficiency reduced placental FABP4 mRNA in rats, which may reflect changed FA handling [131]. Insufficient calcium status in pregnant stage up-regulated FABP4 in liver and adipose tissues in the male offspring [132,133] while maternal VD deficiency increased FABP4 gene expression and exacerbates the dysbiosis of gut microbiota in the gastrointestinal tract of their offspring in mice [134]. Mice with reduced n-3 LCPUFAs intake showed altered placental vascular architecture and increased placental FABP4 expression [135,136]. Given the certain relationships between FABP4 and metabolic syndromes, these alternations may contribute to hepatic lipid accumulation, metabolic disturbances, and obesity in their later life [132,133,134].

## 6. The Potential Clinical Implications of FABP4 in Reproductive and Gestational Health

### 6.1. FABP4 Is a Potential Biomarker for Gestational Complications

Serum FABP4 level has been documented as a specific biomarker for metabolic syndromes [137] and cardiovascular diseases [26]. As aforementioned, correlations were observed between abnormal circulating FABP4 levels and the onset of reproductive and pregnant complications including PCOS, miscarriage, GDM, PE, and fetal development dysfunction (Figure 2). These findings suggest that circulating FABP4 is a potential biomarker for various maternal–fetal diseases. However, the practical value of FABP4 in the clinic is still needed to be validated in large-scale cohorts to determine their cut-off value for specific diseases. Moreover, prospective studies are appealed to examine the predictive value of FABP4 on various pregnancy outcomes.

### 6.2. A Potential Therapeutic Strategy for Pregnancy-Related Disorders by Targeting FABP4

Given the well-known features of FABP4 in metabolic disorders and inflammation, blocking or neutralizing FABP4 by either chemical inhibitors or monoclonal antibodies has emerged as a novel therapeutical strategy. Selective inhibitors of FABP4, such as BMS309403, have been developed to blockade the binding between FABP4 and endogenous FAs [138]. Animal studies showed that BMS309403 treatment effectively ameliorated GDM symptoms in *C57BL/KsJdb^−^/^+^ (db/^+^)* GDM mice model, including improving insulin and glucose metabolism, reducing circulating proinflammatory cytokines, and preventing macrophages infiltration in adipose tissues [139,140]. Furthermore, one in vitro study showed that BMS309403 treatment in trophoblast cells produced fewer fatty acids, which could be hypothesized that in vivo this might reduce macrosomia [14]. These findings suggests that blocking FABP4 is a potential therapeutic approach for gestational and fetal complication deriving from metabolic disorders and inflammation.

However, though excess serum FABP4 concentration was associated with negative outcomes during pregnancy, the essential role of FABP4 in lipids metabolism and placental and fetal development should also be considered. VEGF is a vital factor for placenta development during pregnancy. Interestingly, FABP4 emerged as a downstream target of VEGF [73]. Knockdown of FABP4 in endothelial cells significantly impaired VEGF-mediated angiogenesis and cell proliferation [73]. Meanwhile, FAPB4 is also an important effector of PPARγ activation [141], which is required for placenta development and lack of PPARγ resulted in labyrinth dysfunction and fetal lethality [142]. Leptin treatment stimulated tube formation in both human first-trimester extravillous placental trophoblast cells and *HTR8/SVneo* cell line, which was mainly attributed to increasing placental FABP4 expression rather than other known angiogenic factors [143]. Therefore, these results indicate FABP4 is implicated in promoting pregnancy establishment and embryo development.

Collectively, current evidence shows divergent effects of FABP4 on pregnancy outcome. Whether targeting FABP4 serves as a therapeutic approach to improve pregnancy outcomes should be cautiously viewed. Further research is still required to identify the specific function of FABP4 in different pregnancy conditions. Second, due to the side effects of FABP4 inhibitors on cardiac function [144], the safety of FABP4-based therapy on both maternal and fetal health is warranted to be further verified.

## 7. Summary

FABP4 is a pivotal adipocytokine that serves integral roles in terms of lipid transport, glucose metabolism, and inflammation. However, most previous studies determine FABP4 functions in metabolism in males, with a relative lack of studies focusing on its impact on females. Hence, the current understanding of FABP4 in reproduction and offspring remains obscure. This review explores the physiological implications of FABP4 in reproduction, pregnancy, and its subsequent influence on offspring health.

As a protein expressed predominantly in the endothelium of the placenta and trophoblasts, FABP4 modulates the proliferation, migration, and invasion of endometrial epithelial cells and trophoblast cells. Coupling its role in glucose and lipid metabolism regulation with its impact on localized immunological aberrations, it plays a crucial role in pregnancy establishment and maintenance and further impacts fetal growth and neonatal size. Notably, increased FABP4 serum levels are widely associated with the risk of various pregnancy conditions, including miscarriage, GDM, PE, and premature delivery. Therefore, maintaining FABP4 hemostasis is crucial for pregnancy health. Both excess and deficiency of FABP4 may disturb the meticulous governed physiological dynamic engaged during gestation and lead to pathological implications. These findings also indicate the potential of FABP4 as a biomarker and therapeutical target for women’s reproductive and pregnancy complications, with future research being crucial for their clinical realization. However, there are several significant challenges yet to be addressed in the field of FABP4 research in women reproduction and pregnancy. First, it remains mostly unclear whether elevated FABP4 level is a co-factor or a cause-effect in a pathological process. Despite current evidence, elaborated FABP4 deficient animal models are needed to illustrate the cause-effect of FABP4 and how FABP4 regulates reproduction and gestation health at molecular levels. Second, since current findings about FABP4 are largely derived from non-pregnancy contexts, whether they could be applied in pregnancy should be approached with caution. It would be beneficial to understand the tissue sources and upstream regulators for elevated FABP4 level in pregnant complications. More mechanistic studies into the regulation of FABP4 levels during pregnancy are needed, potentially paving the way for novel therapeutic strategies targeting FABP4 to improve pregnancy outcomes. Moreover, a majority of studies related to FABP4 in pregnancy are observational. Further extensive investigations including large-scale, longitudinal clinical trials are paramount to justify the diagnostic and therapeutic potential of FABP4 in maternal–fetal metabolic diseases alongside rigorous safety assessments of its therapeutic application.

## Figures and Tables

**Figure 1 ijms-24-12655-f001:**
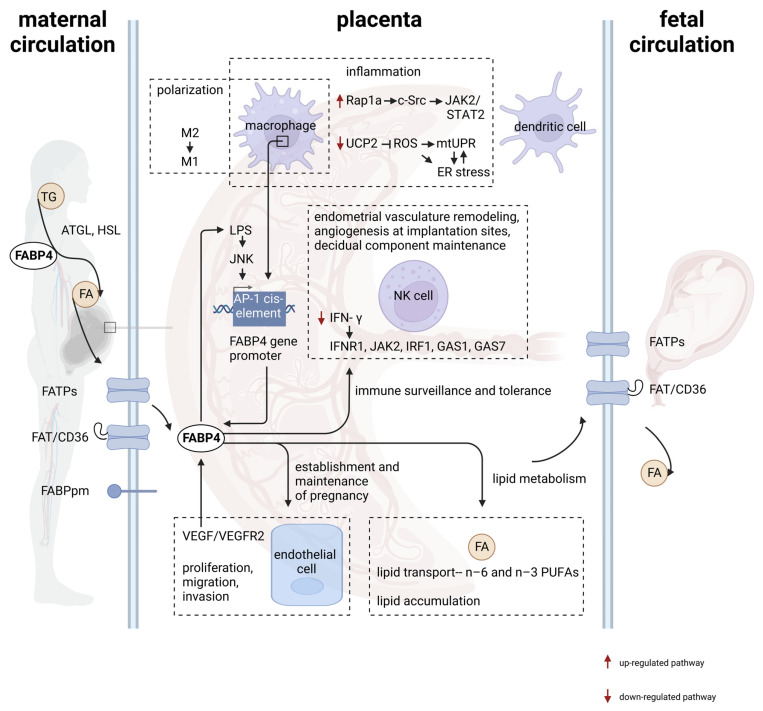
The diverse role of FABP4 in pregnancy health by regulating immunological and metabolic hemostasis in the placenta. FABP4 is expressed in immune cells (including macrophage, NK cells, and dendritic cells) and endothelial cells in the placenta. In placenta-resident macrophages, increased FABP4 enhances inflammation through (1) inducing a shift from the M2 anti-inflammatory phenotype to M1 pro-inflammatory phenotype; (2) activating the JNK pathway, (3) triggering ROS production, mtUPR, and ER stress [68]. FABP4 expression also impaired the production of IFN-γin NK cells, a vital cytokine for pregnancy success. Within endothelial cells, FABP4 is upregulated via VEGF/VEGFR2 pathway [73], facilitating cell proliferation, migration, and invasion. In terms of lipid metabolism, FABP4 metabolizes TG into FFAs by targeting HSL and ATGL. The FFAs are transported from the maternal circulation into the parental circulation via the placenta through FATPs and FAT/CD36 [74]. Additionally, FABP4 also facilities placental lipid transport (n-6 and n-3 PUFAs) and accumulation at the maternal–placenta interface and endothelial layer. FABP4, adipocyte fatty acid-binding protein; NK cells, natural killer cells; JNK, c-Jun N-terminal kinase; ROS, reactive oxygen species; mtUPR, mitochondrial unfolded-protein response; ER, endoplasmic reticulum; IFN-γ, interferon-γ; VEGF, vascular endothelial growth factor; VEGFR2, vascular endothelial growth factor receptor 2; TG, circulating triglyceride; FFAs, free fatty acid; FATPs, fatty acid transport proteins; FAT/CD36, fatty acid translocase; PUFAs, polyunsaturated fatty acids.

**Figure 2 ijms-24-12655-f002:**
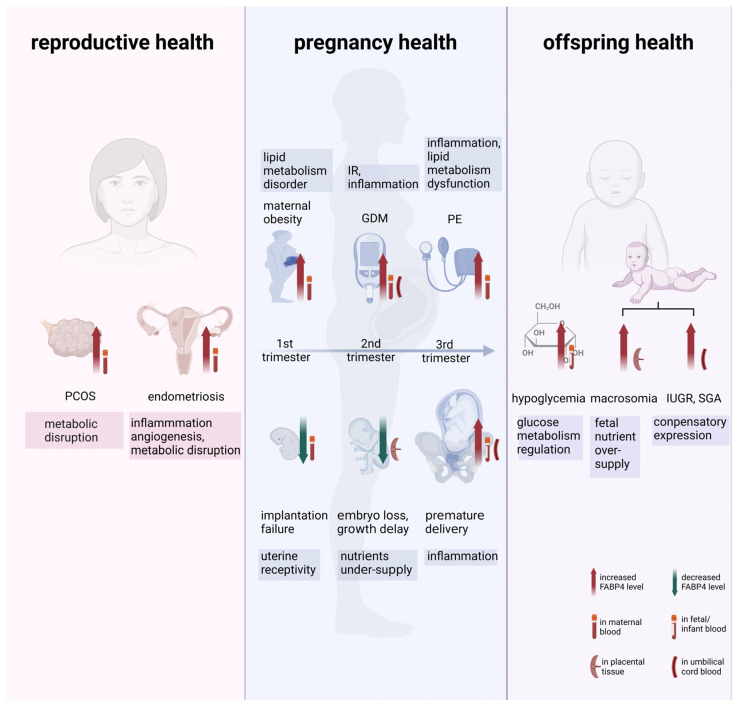
The impact of altered FABP4 levels on female reproduction, pregnancy outcomes, and offspring health. Alternations in FABP4 levels are observed in various diseases. Increased FABP4 levels are observed in patients with endometriosis and PCOS. During pregnancy, FABP4 levels fluctuate throughout different gestation stages. High maternal FABP4 levels have been associated with a risk of miscarriage, maternal obesity, GDM, PE, and premature delivery. However, deficient FABP4 in the mid-gestation can lead to nutrient under-supply and placental/decidual growth impairments. In neonates, FABP4 increases in hypoglycemia cases and exhibits a U-shaped correlation with neonatal size. The red and green arrow means increased or decreased FABP4 levels, respectively. FABP4, adipocyte fatty acid-binding protein; PCOS, polycystic ovary syndrome; GDM, gestational diabetes mellitus; PE, preeclampsia.

## Data Availability

No data is contained within the article. Additional information is available on request from the corresponding author.

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
