# Peer review of "Pathophysiological Insight into Fatty Acid-Binding Protein-4: Multifaced Roles in Reproduction, Pregnancy, and Offspring Health"

_ijms, 2023, doi:10.3390/ijms241612655_

Round 1
Reviewer 1 Report
GENERAL REMARK:
This is a useful overview of the role of FABP4 in reproduction and pregnancy. As with most studies that target on one substance, it remains mostly unclear if elevated FABP4 level is a co-factor in a pathological process, or if it is really a step in the causal pathway.
In several paragraphs it should be written more explicitly if findings were derived from human, animal or in vitro experiments.
SPECIFIC REMARKS:
Line 177 – 184: In line 177-182 you write that absence ofFABP4 has no effect on the fetal development in mice. However, in line 183 you write that FABP4 plays a crucial role in pregnancy.
Table 1: what is a “distinguishing capacity” – you mean RR or OR? In some studies you specify that blood was taken after fasting, but in others not – I would expect that blood was always taken after fasting – is this correct? Possibly you could make the description of the studies more uniform. Now you write sometimes that FABP$ was higher in GDM than in controls, and for other studies that it was associated with GDM. You might consider to add a column specifying if levels were significantly higher in case vs controls.
Line 239: “Importantly, FABP4 may also be important” – better remove the first word.
Line 302-304: If FABP4 is useful for assessment of risk of GDM depends on the costs and availability of the test. For monitoring treatment other tests are more useful like monitoring blood glucose and HbA1c.
Line 321-323: FABP4 as an early biomarker for PE: any data on the association of elevated FABP4 and other biomarkers? Risk factors for PE as obesitas or hypertension are associated with elevated FABP4, so the question is if FABP4 is independent from other risk factors as a biomarker.
Line 331” Increased serum FABP4 levels have been reported in premature delivery” – please add “in one study and change increased to higher.
Line 334-335 “tends to increase in preterm neonates, especially in those with a smaller gestational age.” – I suppose you want to say that in preterm neonates FABP4 tends to be higher in cord blood and at 1 month of age than in neonates delivered at term, and this difference increases with a lower gestational age at delivery. Note: increase is a dynamic process, while higher only specifies a difference.
Line 339 “Therefore, there was a positive correlation between serum FABP4 levels and the onset of preterm labor” - better remove this sentence, or change “was” to “could be”, in table 1 you show 2 papers, one shows a difference and the other does not.
Line 343: “Fetal glucose production is suppressed during pregnancy” – I doubt if this is true and ref 106 and 107 do not support this statement. 107 describes glucose levels postpartum. Or do you mean shortly after delivery, but then it is mainly intake reduction.
Line 362: “Low birth weight is defined as less than 2500g weight at birth” – this is a rather old fashioned definition. Currently, it is more common to use percentiles.
Line 427-430: This sentence is a bit misleading. The paper only shows that trophoblast cells in culture produce less fatty acids after blocking FABP4 and if this ultimately reduces macrosomia remains to be seen . You should rephrase this sentence like “One in vitro study showed that trophoblast cells produced less fatty acids after blocking FABP4. It could be hypothesized that in vivo this might reduce macrosomia.”
See above
Reviewer 2 Report
The review by Shi et al. provides a comprehensive and up-to-date analysis of the available literature on FABP4 and its relevance to pregnancy outcomes. However, to further enhance the significance of the review, the following suggestions are proposed:
- Inclusion of Tissue Source and Expression Levels: To gain a deeper understanding of the mechanisms behind increased plasma levels of FABP4, it would be beneficial to discuss the tissue source responsible for its production and secretion. Additionally, exploring the levels of FABP4 expression in different tissues and the secretion patterns could provide valuable insights. For instance, addressing whether the increased plasma levels are primarily due to enhanced secretion by specific tissues and whether this leads to FABP4 deficiency in tissues/cells.
- Clarification on FABP3 Mention: On page 4, line 167, there is an unexpected mention of FABP3, which could potentially confuse readers. It is recommended to either provide a brief explanation of the relevance of FABP3 in this context or remove the reference to maintain clarity and focus in the discussion.
- Causality of Increased FABP4 in Pathological Pregnancies: To establish a stronger argument regarding the significance of increased FABP4 in pathological pregnancies, it is essential to address whether this increase is a cause or a consequence. Presenting relevant evidence and discussing the implications of elevated FABP4 levels in pathological conditions will contribute to a more comprehensive understanding.
- Dedicated Section on Factors Triggering FABP4 Expression: Considering the authors have touched upon potential triggers of increased FABP4 expression and levels throughout the manuscript, it would be advantageous to include a dedicated section that thoroughly explores these factors. For instance, highlighting the role of endocrine hormones like testosterone, insulin, and leptin in triggering FABP4 expression and plasma level changes can offer valuable insights into the regulatory mechanisms.
- Concluding Remarks on FABP4 Levels: In the conclusion section, it would be prudent to emphasize that maintaining normal levels of FABP4 is crucial for healthy pregnancy outcomes. Both excess and deficiency of FABP4 may have pathological implications. This will provide a clear takeaway message for readers and highlight the clinical significance of FABP4 regulation during pregnancy.
